# FP-Rainbow : Fingerprint-based Browser Configuration Identification

## ABSTRACT

Browser fingerprinting is a tracking technique that collects attributes and calls functions from the browser's APIs. Unlike cookies, browser fingerprints are difficult to evade or delete, raising significant privacy concerns for users as they can be used to re-identify individuals over browsing sessions without their consent. Yet, there has been limited research on the impact of browser configuration settings on these fingerprints.

This paper introduces FP-Rainbow, a novel approach to systematically explore and map the configuration space of Chromium-based web browsers aiming to identify the impact of configuration parameters on browser fingerprints and their changes over time. We explore $1,748$ configuration parameters (switches) and identify their impact on the browser's BOM (Browser Object Model). By collecting and analyzing over $61,000$ fingerprints from 18 versions of Chromium, our study reveals that 32 to 56 of these configuration parameters (depending on versions), such as `disable-3d-apis` or `disable-notifications`, influence the fingerprint of a web browser.

FP-Rainbow also proves efficient in identifying browser configuration parameters from unknown fingerprints, achieving an average successful identification rate of 84% when considering a single configuration parameter and 78% when multiple parameters are involved, across all evaluated browser versions. These findings emphasize the importance of measuring the impact of configuration parameters on browsers to develop safer and more privacy-friendly web browsers.

## CCS CONCEPTS

• **Security and privacy** → **Web application security**; • **Information systems** → *Browsers*.

## KEYWORDS

Browser Fingerprinting, Privacy, Web Security, Online Tracking, Configuration Parameters, BOM Exploration

**ACM Reference Format:**
Anonymous Author(s). 2024. FP-Rainbow : Fingerprint-based Browser Configuration Identification. In *Proceedings of Make sure to enter the correct conference title from your rights confirmation emai (Conference acronym 'XX).* ACM, New York, NY, USA, 9 pages. https://doi.org/XXXXXXX.XXXXXXX

## 1 INTRODUCTION

Web browsers, now used by billions of people, constantly evolve to meet the Internet's growth and user expectations. They incorporate new features, allowing developers to deploy more sophisticated apps with each version. Simultaneously, users can personalize their browsing experience through the various settings of their browsers. A Europe-wide study shows that 36% of users reported having modified their browser settings to prevent or limit the use of cookies [1]. Combined with the popularity of many browser extensions, many users personalize their browsing experience. This, however, leads to browser configurations that are potentially detectable due to side effects, some configurations may change the performance or behavior of a browser, while others may affect the *Browser Object Model* (BOM) and JavaScript APIs exposed by the browser, making them thus detectable. The BOM is distinct from the *Document Object Model* (DOM). While the DOM focuses on the content of a web page, the BOM is a programming interface that enables interaction with the browser. It provides access to objects, attributes, properties, and methods associated with the browser window. This makes it possible to retrieve more information about the browser's environment and configuration. These side-effects leak information to attackers that can be used to identify exploitable configurations (e.g., unsafe or experimental settings) or to simply increase the uniqueness of browser fingerprints and improve tracking algorithms.

Browser fingerprinting is a well-known technique to re-identify browsers [1, 14, 19]. Most fingerprinting algorithms use a few dozen attributes to create a unique identifier. These attributes are chosen because of their entropy, which provides uniqueness, and their stability between executions. Common attributes include the user agent, language, screen resolution, time zone, and plugins, as well as more complex attributes, such as canvas images to fingerprint the graphics layers [17, 21], font enumeration to recover installed fonts [9, 23], browser extension fingerprinting [30, 35], and GPU [16] or CPU fingerprinting [27, 31]. However, it has been shown that the browser exposes a much larger set of attributes that can potentially be exploited by attackers [29]. By enumerating the *Browser Object Model* (BOM), starting from the `window` object, we find anywhere from 12 to over 16 thousand objects, attributes, and methods exposed. Furthermore, little is known about how a browser's configuration affects the BOM, nor about the side-effects browser developers introduce when developing new features. To the best of our knowledge, no work has explored the impact of browser configuration settings on browser fingerprints. This information is useful in different scenarios, for both developers but also to attackers. For example, developers need information about the client's environment and configuration parameters to better reproduce their environment and quickly fix bugs. For attackers, they can identify configurations that are known to have security issues and exploit them.

---

[1] https://ec.europa.eu/eurostat/web/products-eurostat-news/-/edn-20220208-1

In this paper, we present FP-Rainbow, an approach to systematically explore and identify browser settings and configurations that affect the *Browser Object Model* (BOM) and the browser's fingerprint. For every configuration parameter available, we execute the browser to enumerate its BOM and collect an extensive fingerprint of around 15 thousand attributes depending on the browser's version and configuration. We then analyze the fingerprints to identify which configuration parameters impact which fingerprint attributes, if any. With this information, we provide a method to reidentify configuration parameters from the browser's fingerprint.

The key contributions of our work are:

- A dataset consisting of an exploration of 1, 748 configuration parameters in 18 versions of Chromium, for which we collected 61, 559 browser fingerprints, each containing between 12, 334 and 16, 473 attributes.
- The identification of 32 to 56 configuration parameters, depending on the browser's version, that impact the BOM. We also document the attributes that are added, removed or changed for each configuration parameter across different browser versions, finding relatively few collisions between switches in terms of impacted attributes.
- Through 1, 116 randomly sampled browser configurations, we show that a dataset calculated on a single device can be leveraged to successfully identify configuration parameters from other devices.
- A method for (re-)identifying browser configuration parameters from a browser's fingerprint by comparing attribute subsets, achieving an average successful identification rate of 84, 36% across all browser versions and configuration parameters that impact the BOM.

The remainder of this paper is structured as follows. Section 2 provides an overview of the current state of the art in the field of browser fingerprinting and motivates our work. Section 3 introduces FP-Rainbow. Section 4 outlines our experimental methodology, while Section 5 presents the results of our experiments. Section 6 concludes the paper.

## 2 RELATED WORK

Given the increasing awareness of privacy concerns [10], research has been undertaken to strengthen user protection while browsing [24, 26, 32]. In the last decade, researchers and privacy advocates have thus explored various aspects of browser fingerprinting and its implications for user privacy [1, 3, 8, 35]. A substantial body of related work has shed light on the prevalence, mechanisms, and challenges associated with browser fingerprinting [7]. Some key findings and approaches from prior research include:

*Fingerprinting Techniques and Evasion.* In the past few years, many studies have contributed valuable insights to the field of fingerprinting [8, 14, 22]. Research by Acar *et al.* [1] and Englehardt *et al.* [8] highlighted the sophistication of browser fingerprinting techniques employed by online entities. These techniques combine attributes to create a unique fingerprint, enabling user tracking across different sites, even when attempting anonymity through tools like VPNs or incognito mode. Unlike traditional methods such as cookies, browser fingerprinting is challenging for users to control or detect. Users cannot disable fingerprinting without breaking their Web experience due to the very nature of how modern browsers work. These studies demonstrated the ability to track users across websites and identified various attributes used in fingerprinting [19, 23, 34]. They also investigated techniques to evade fingerprinting and the effectiveness of privacy-enhancing browser extensions.

*Anti-Fingerprinting Tools.* Efforts to counter browser fingerprinting have resulted in the development of diverse anti-fingerprinting methods and techniques[28]. One notable solution proposed by Laperdrix *et al.* [18] aims to mitigate the risks associated with browser fingerprinting by constantly randomizing parts of the browser's environment that influences the browser's fingerprint, within a controlled environment. In a similar vein, Azad *et al.* [4] conducted tests on various tools designed to spoof the browser fingerprint and discovered that certain attributes can be exploited to detect anti-fingerprinting tools. These anti-fingerprinting tools are primarily focused on manipulating the attributes utilized in fingerprinting to reduce the uniqueness of user fingerprints and bolster privacy. By altering and obfuscating specific attributes, these tools seek to make it more challenging for online entities to track and identify users based on their browser fingerprints. However, these tools can impact the user experience and break some websites. They can also be recognizable and are difficult to maintain over time [33].

*Configuration Exploration.* As presented in the survey proposed by Pereira *et al.* [25], many studies focus on exploring configurations, *e.g.*, for performance prediction [12], optimization [11] or dynamic reconfiguration [20]. A wide variety of domains have been studied, such as the Linux kernel [2], JVM parameters [6], or video encoding [15]. Huyghe *et al.* [13] generate fingerprints by running browser configurations, arguably the closest work to ours, but they do not explore the impact the configurations have on the browser's fingerprints, nor identify which configuration parameters are mapped to specific parts of the fingerprint.

*Motivation.* Researchers have explored the impact of changes in browsers and Web standards on fingerprinting resistance [5]. Various studies have examined the potential of browser configurations, their code [26], and standard updates to reduce the effectiveness of browser fingerprinting techniques. Despite progress in understanding and addressing browser fingerprinting, several limitations persist in existing approaches like the browser's complexity, its wide range of uses and its tendency to evolve rapidly.There is thus a critical need to thoroughly explore browser configurations to pinpoint configurations and settings that may pose privacy or security risks, as well as provide tools to assist browser developers in minimizing the side-effects of their code on the Browser Object Model (BOM).

## 3 FP-RAINBOW

We present FP-Rainbow, our approach to systematically explore browser configurations to identify the impact they have on the Browser Object Model (BOM) through the collection of exhaustive fingerprints. Our objective is to identify how each configuration parameter impacts the BOM and thus reveals information about the browser's configuration.

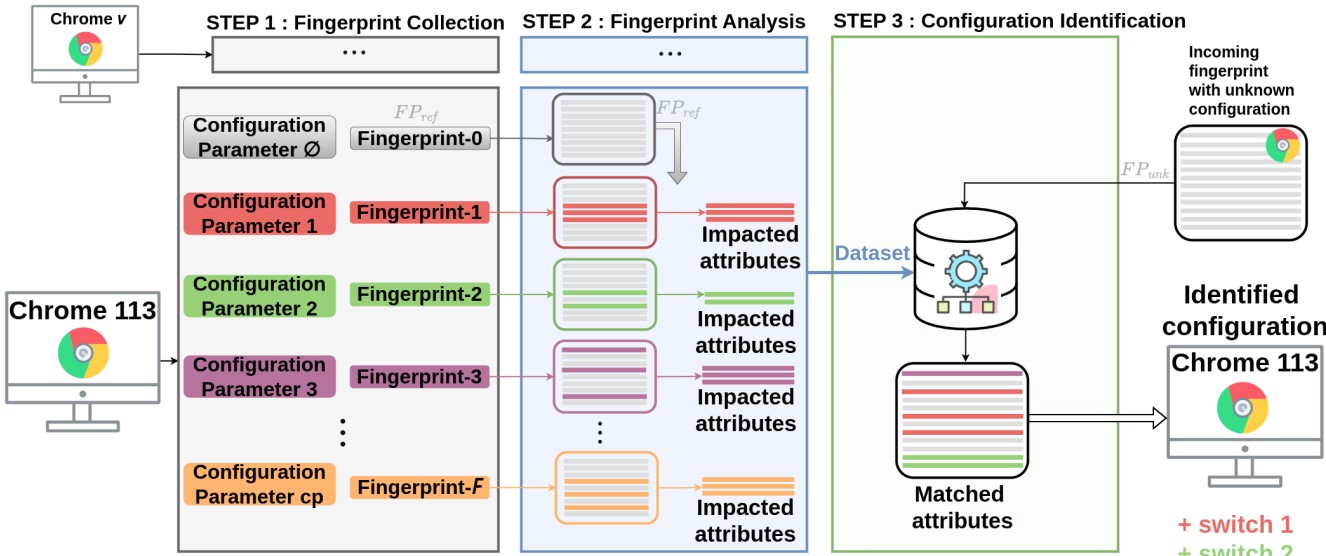

Figure 1: Overview of the FP-Rainbow approach

FP-Rainbow relies on a three-step process, as depicted by Figure 1. First, fingerprints are collected from different versions of Chromium. More precisely, one fingerprint is generated for each configuration parameter of each browser version. Second, each fingerprint is compared to the default browser configuration to determine which attributes are impacted by the configuration parameter. Many configuration parameters do not impact the BOM. We are interested in those that do. We use the knowledge of which configuration parameters impact what attributes of the BOM for version of the browser to:

- Given a fingerprint from a browser with an unknown configuration, we identify as many configuration parameters as possible that the browser has enabled.
- Calculate which attributes are good indicators of the activation of a configuration parameter by weighting their importance using a stability metric we develop.

In our current implementation, FP-Rainbow targets one type of configuration parameter, namely command-line switches[2] but is extensible to flags and other configuration options.

## 3.1 Fingerprint Collection and Analysis

**Fingerprint Collection**. FP-Rainbow collects fingerprints and the browser's configuration, including the operating system it runs on, browser's version, and various configuration parameters. Specifically, FP-Rainbow systematically executes multiple instances of the browser, each with distinct versions and configuration parameters, and retrieves the resulting fingerprint. Such a fingerprint and its related browser configuration are then stored in a database to be further analyzed (see next steps). To delve into the details of the BOM, FP-Rainbow builds upon and extends two existing

techniques, namely FingerprintJS[3], a popular open-source browser fingerprinting tool specialized in re-identifying browser's for tracking or authentication services, and JavaScript Template Attacks[4], a BOM enumeration solution developed by Schwarz *et al.* [29].

**Fingerprint Analysis**. Each generated fingerprint $FP_{new}$ is compared with the reference fingerprint $FP_{ref}$, *i.e.*, the fingerprint devoid of any manually set configuration parameters, retaining only the default configuration parameters. This comparison aims to discern any distinctions introduced by the adjustment of configuration parameters. During the comparison, FP-Rainbow focuses on attribute differences between each $FP_{new}$ and $FP_{ref}$. In particular, for every attribute in the fingerprint $FP_{new}$, FP-Rainbow distinguishes between four cases:

- The attribute exists in $FP_{ref}$ with the same value.
- The attribute exists in $FP_{ref}$, but its value differs.
- The attribute does not exist in $FP_{ref}$.
- The attribute does not exist in $FP_{new}$.

FP-Rainbow associates each configuration parameter with the fingerprint it generates and identifies the impact on the browser's BOM. The objective is twofold: to help identify the attributes impacted by different configuration parameters and to build a knowledge base that can be leveraged to identify the configurations from unknown fingerprints. In short, we consider :

- $FP_{ref}$: The referent fingerprint which serves as the baseline for a specific browser version (one per version).
- $FP_{new}$: The fingerprint generated by applying a specific configuration parameter (one per parameter per version).
- $FP_{unk}$: The unknown fingerprint, where we aim to determine the configuration parameters used to generate it.

---

[2]https://chromium.googlesource.com/chromium/src/+/main/docs/configuration.md#Switches

[3]https://github.com/fingerprintjs/fingerprintjs
[4]https://github.com/IAIK/jstemplate

**Configuration Identification**. Relying on the dataset of impacted attributes, FP-Rainbow can attempts to identify configuration parameters from a browser by analyzing its fingerprint. To do so, for each configuration parameter from the unknown fingerprint $FP_{unk}$, attributes known to be impacted by a configuration parameter (from the previous fingerprint analysis step, *e.g.*, by a changed switch) are extracted and assessed, falling into one of the three following cases:

- Added: The unknown fingerprint $FP_{unk}$ has the attribute known to be impacted by the configuration parameter.
- Changed: The unknown fingerprint $FP_{unk}$ has the attribute known to be impacted by the configuration parameter but with a different value.
- Removed: The unknown fingerprint $FP_{unk}$ does not have the attribute known to be impacted by the configuration parameter.

**Extracting subsets of attributes per configuration parameter**. To identify an unknown fingerprint's configuration parameters, we compare it with the impact analysis for each configuration parameter from our dataset. The set of impacted attributes for each configuration parameter is small compared to the entirety of the fingerprint. We do not compare the fingerprints entirely since this will simply pollute the calculation, especially if multiple configuration parameters are activated or if the fingerprint originates from different environments. During the fingerprint analysis, as described previously, we identify the impact of each configuration parameter on the browser's fingerprint, in terms of attributes added, changed, or removed attributes. This provides us with a subset of attributes that are known to be impacted by the configuration parameter. We then extract the same attributes of the subset from the unknown fingerprint $FP_{unk}$ to compare them. Similar to the analysis step, each attribute may exist, not exist, or exist with a different value. We then compare the extracted set of attributes from the unknown fingerprint $FP_{unk}$ to those from the configuration parameter and decide if the parameter is active or not in the unknown fingerprint $FP_{unk}$. Given the amount of variability found in browser fingerprints, we cannot expect our comparisons to always be identical.

## 3.2 Stability Analysis of Impacted Attributes

During fingerprint generation, we collected between $12,334$ and $16,473$ attributes per fingerprint. Many attributes can vary between devices or across different browser versions. Additionally, a single configuration parameter may modify multiple attributes, or certain attributes might no longer be impacted in newer browser versions. To address this, we focus on identifying stable attributes that remain consistent in detecting configuration parameters. We introduce the concept of attribute stability and use it to quantitatively evaluate how stable specific attributes are across various browser versions under varying configuration parameters. An attribute is considered perfectly stable when it is influenced by only a single configuration parameter, regardless of the browser version.

Let $CP$ be the set of configuration parameters, $A$ the set of attributes, and $V$ the set of browser versions. Then, $cp$ is a configuration parameter from fingerprint $FP_{new}$ such as $cp \in CP$. $cp_{default}$ represents the default configuration used in $FP_{ref}$. For each version $v \in V$, each attribute $a \in A$ is either associated with a version $v$ and a configuration parameter $cp$, and is referred to as $a_{cp,v}$, or with a version $v$ and no configuration parameter for the default configuration $cp_{default}$ and is defined as $a_{cp_{default},v}$. The difference between fingerprints $FP_{ref}$ and $FP_{new}$ for each attribute $a$ is computed as follows:

$$Impacted_{a_{cp,v}} = 1 \text{ if and only if } a_{cp,v} \neq a_{cp_{default},v} \text{ (0 otherwise)} \tag{1}$$

Thus, $Impacted_{a_{cp,v}} = 1$ if the attribute $a$ is impacted (changed, added, or removed). Else $Impacted_{a_{cp,v}} = 0$ if the attribute is not impacted.

We evaluate stability across all configuration parameters $cp \in CP$. The more configuration parameters impact a given attribute, the more unstable the attribute is. If an attribute is impacted by only one configuration parameter, we consider it perfectly stable. To determine the stability of all attributes $a \in A$ for each configuration parameter $cp \in CP$, we check if an attribute $a$ is impacted by all configuration parameters $cp$ relying on the following formula:

$$Stability_{a,v} = \left(1 - \left(\frac{\sum_{cp \in CP_v} Impacted_{a_{cp,v}}}{|CP_v|}\right)\right) \tag{2}$$

Where $|CP_v|$ is the total number of configuration parameters that have impacted an attribute $a$ for a version $v$. Thus, $Stability_{a,v} = 1$ if and only if the attribute $a$ is impacted by only one configuration parameter $cp \in CP$. That is, $0 \leq Stability_{a,v} \leq 1$: the more configuration parameters impact a given attribute, the lower $Stability$ while a value close to 1 signifies stability for attribute $a$ across all configuration parameter $cp \in CP$ for a version $v \in V$.

To compute the stability of an attribute across all browser versions as well as to consider other environments or browser variations such as Headless and Headful, Equation (2) is iteratively applied over all versions $v \in V$. The result is then divided by the total number of browser versions $|V|$ as follows:

$$\omega_a = \frac{\sum_{v \in V} Stability_{a,v}}{|V|} \tag{3}$$

The overall stability of an attribute, $\omega_a$, is thus computed such that $0 \leq \omega_a \leq 1$, with an attribute $a$ being more stable as its related $\omega_a$ gets closer to 1.

Relying on Equation (3), one can measure the importance and stability of browser attributes across various versions and configuration parameters ($\approx 1,748$ switches). An attribute with a value $\omega_a$ close to 1 is highly important, stable, and reliable for use when identifying configuration parameters. Conversely, a value close to 0 indicates high variability and less reliability. This approach identifies essential attributes, minimizing time and resources for generating browser fingerprints while maintaining efficiency and accuracy in identifying configuration parameters.

## 3.3 Fingerprint Comparison for Configuration Identification

To identify a configuration parameter, we compare subsets of impacted attributes that we retrieved using $impacted_{a_{cp,v}}$ in equation 1. We define the set of impacted attributes resulting from our formula 1 as *impactedSet* for all versions, and *impactedSubset*$_v$ for a specific

version, such that $impactedSubset_v \subseteq impactedSet$. During identification, we select only the attributes of the unknown fingerprint $FP_{unk}$ that exist in our set $impactedSet$. By definition, the set of selected attributes from the unknown fingerprint $FP_{unk}$ is less than or equal to the set of attributes in $impactedSet$. We have categorized the impacted attributes into three categories:

- $Add_{a,impactedSubset_v,unk} = 1$ if and only if the attribute $a$ didn't exist in fingerprint $FP_{ref,v}$ but exists in $impactedSubset_v$ and in $FP_{unk}$,
- $Chg_{a,impactedSubset_v,unk} = 1$ if and only if the value of attribute $a$ is different in fingerprint $FP_{ref,v}$ and $impactedSubset_v$ but is identical between $impactedSubset_v$ and $FP_{unk}$,
- $Rm_{a,impactedSubset_v,unk} = 1$ if and only if the attribute $a$ exists in fingerprint $FP_{ref,v}$ but does not exist in $impactedSubset_v$ nor in $FP_{unk}$.

For each attribute, we have three cases, as indicated above (i.e., Added, Changed, Removed). We count the occurrences of each case and normalize the value between 0 (the configuration parameter does not match) and 1 (the configuration parameter matches perfectly). To simplify we make the sum of the three case such as :

$$IdentifiedSet_v = Add_{a,impactedSubset_v,unk}$$
$$+ Chg_{a,impactedSubset_v,unk} \quad (4)$$
$$+ Rm_{a,impactedSubset_v,unk}$$

Then, we define:

$$Similarity_{(impactedSubset_v),(FP_{unk})} =$$
$$\frac{\sum_{a \in A_{impactedSubset_v}} (IdentifiedSet_v)}{|A_{impactedSubset_v}|} \quad (5)$$

Where $|A_{impactedSubset_v}|$ is the number of impacted attribute $a$ for a version $v$. From this equation, we can identify the configuration parameters $cp$ used by an unknown web browser based on its fingerprint $FP_{unk}$ by comparing each impacted attribute between the subset $impactedSubset_v$ and the unknown fingerprint.

The behavior of the $Chg_{a,impactedSubset_v,unk}$ operator is adapted for each type of data being processed, such as strings, numerical values, or PNG images. For numerical values, a simple absolute difference is often sufficient. However, to prevent large differences from skewing the comparisons, we assign normalize the differences. For strings, differences are measured using the Levenshtein distance. Regarding PNG images, particularly in the context of comparing Canvas fingerprints [21], the sum of the absolute pixel-by-pixel differences is generally sufficient. Given the large image size and the number of channels (i.e., RGBA), this absolute difference is normalized similarly to numerical values. By summing the pixel-by-pixel differences, it reduces to a single numerical value, allowing PNG images to be processed and normalized as numerical parameters.

## 4 METHODOLOGY

This section describes our hardware settings, our data collection process, and the comparison methods we employed. All data gathered during our experiments are publicly accessible [5].

[5]https://github.com/fp-rainbow/fp-rainbow

### 4.1 Research Questions

In this paper, we propose an approach to systematically explore and identify browser settings and configurations that affect the BOM, and aim to answer the following research questions:

**RQ1**: *What impact do individual browser configurations parameters have on the BOM?* In particular, we want to assess the impact of one type of configuration parameters: switches.

**RQ2**: *Is it possible to identify a configuration parameter from an unknown fingerprint?* Comparing an unknown fingerprint to a set of known-to-be-impacted configuration parameters can provide insights into the unknown browser configuration.

**RQ3**: *Can a dataset generated from a single device effectively be leveraged to identify fingerprints across multiple devices?* If so, this could reduce the need for generating datasets on many, diverse devices, and instead, for example, allow calculating the dataset once on a more powerful server.

### 4.2 Software Setup and Hardware Settings

To isolate all components and enforce reproducibility, we conducted all our experiments using Docker and Docker Compose. Each software component, including the web browser, the database, the analysis tools, and the web server, is thus encapsulated in a Docker container, ensuring portability and reproducibility across different environments. Each container has its own Puppeteer[6] instance, and the same database is shared across all Docker containers. All experiments were conducted on a machine featuring an : 2x Intel(R) Xeon(R) Gold 5118 CPU and 188GB of RAM, operating on Rocky Linux with kernel version 4.18 for the dataset, and a Laptop with an Intel(R) i7-1185G7 CPU and 64GB of RAM, operating on Arch Linux with kernel version 6.11 for the generation of unknown browser fingerprints.

### 4.3 Experimental Protocol

*4.3.1 Fingerprint Collection.* We start Chromium browser with Puppeteer and inject configuration parameters into the browser's configuration. We run our browsers in a dedicated Docker container to facilitate reproduction and deployment. The configuration parameters are fetched from the database, a new entry is generated in the database with a UUID. The browser accesses our fingerprinting webserver that collects the fingerprint and cleans unstable attributes (for example, random unique identifiers and hashes that change at every execution). The database stores the browser version, browser configuration parameters, operating system, and CPU architecture, as well as the browser fingerprints. Browser fingerprints are stored in JSON files with unique identifiers (UUIDs) for quick retrieval. In case of a browser crash or unresponsiveness, the latest database entry is marked as `timeout`, this is a rare occurence. We launch one to four browser versions per thread, each with a distinct configuration. The collection is repeated for each configuration parameter $cp \in CP$ available in the database for each version of the browser. The initial launch for each version collects its default configuration, which serves as the reference fingerprint $FP_{ref}$ during the subsequent analysis phase.

The fingerprint for each browser configuration is collected twice, in a row. The attributes that have been changed between the two

[6]https://pptr.dev/

fingerprints are discarded, stabilizing the fingerprint. Overall, $1,748$ switch configuration parameters have been successfully tested. Executing all configurations takes approximately 5 hours on our hardware configuration, as we test up to 4 browser versions per CPU thread. Once all fingerprints and their corresponding browser configurations are collected, the analysis can be started.

*4.3.2 Fingerprint Analysis.* Certain attributes, including the test's URL, test timing, or network type, are not taken into account because they are specific to our test website or our unstable but not caught by our initial double fingerprint filter. Each remaining attribute of every fingerprint $FP_{new}$ is compared with the reference fingerprint $FP_{ref}$, which uses the default configuration. When a difference is detected, the UUID of the fingerprint, the browser configuration, and the observed differences are stored. By running the browser on the same platform, with the same configuration with the exception of one configuration parameter, we isolate the impact of the parameter. Throughout this step, we analyzed over $61,559$ fingerprints with 12 to 16 thousand attributes. After analyzing all versions, the resulting dataset captures the relationship between each configuration parameter and its impact on the BOM for each browser version. This dataset can then serve as a baseline to identify unknown fingerprints.

*4.3.3 Configuration Identification.* To compare an unknown fingerprint $FP_{unk}$ configuration parameters from the impacted attributes, we employed two strategies: one using the fingerprints generated from our BOM exploration and the other using FingerprintJS. In the former case, the fingerprint size can increase as the complete BOM is explored and each attribute-value pair is retrieved. As shown in Table 1, the number of attributes during our experimentation varied between $12,334$ and $16,473$. We selected the known impacted attributes from each switch in our dataset and compare these attributes from the unknown fingerprint. As explained in Section 3, attributes are either added, changed, or have their values removed. Our identification process thus distinguishes between these three cases where we apply our Equation (5). In the case of the FingerprintJS library, a straightforward diff proved sufficient for all three cases given the fingerprint's relatively simple structure and the key/value pair similarity. These two approaches have allowed us to gain a deep understanding of the configuration parameters impact on the fingerprint. We have applied these techniques to all of our datasets.

## 5 RESULTS

This section presents the results obtained through the utilization of our approach, FP-RAINBOW. We first discuss the fingerprint collection and analysis phases. We then delve into the comparison phase and provide insights on configuration parameters that impacted an unknown fingerprint.

## 5.1 Impact of Browser Configurations on the BOM [RQ1]

We generated fingerprints for 18 Chrome browser versions spanning over 18 months, for both the Headful and Headless variants, for a total of 36 browsers. The list of considered versions is available in Table 1. This deliberate choice of a small sample of browser

**Table 1: Analysis of the effects of switches on the BOM, including the generation of browser fingerprints and the number of attribute for specific browser versions**

| Browser version | Number of switches that impact the BOM | Number of switches that impact FingerprintJS | Total Number of switches impacted | Fingerprint Generated | Min attributes per Fingerprint | Max Attributes per Fingerprint |
|---|---|---|---|---|---|---|
| Chrome-113.0.5672.63 | 45 | 15 | 50 | 1713 | 12894 | 15888 |
| Chrome-114.0.5735.133 | 46 | 15 | 50 | 1713 | 12910 | 15917 |
| Chrome-115.0.5790.170 | 46 | 15 | 50 | 1713 | 13014 | 16018 |
| Chrome-116.0.5845.96 | 45 | 15 | 49 | 1713 | 13036 | 16029 |
| Chrome-117.0.5938.149 | 45 | 17 | 49 | 1712 | 13062 | 16031 |
| Chrome-118.0.5993.70 | 44 | 18 | 48 | 1712 | 13072 | 16080 |
| Chrome-119.0.6045.105 | 44 | 18 | 48 | 1712 | 13070 | 16101 |
| Chrome-120.0.6099.109 | 42 | 18 | 47 | 1712 | 13100 | 15945 |
| Chrome-121.0.6167.184 | 42 | 17 | 46 | 1712 | 13120 | 15984 |
| Chrome-122.0.6261.128 | 41 | 17 | 46 | 1712 | 13119 | 16052 |
| Chrome-123.0.6312.122 | 42 | 17 | 45 | 1711 | 13164 | 16128 |
| Chrome-124.0.6367.207 | 41 | 17 | 46 | 1710 | 13189 | 16134 |
| Chrome-125.0.6422.141 | 40 | 16 | 44 | 1710 | 13300 | 16180 |
| Chrome-126.0.6478.182 | 40 | 16 | 44 | 1710 | 13316 | 16227 |
| Chrome-127.0.6533.119 | 40 | 16 | 44 | 1710 | 13335 | 16253 |
| Chrome-128.0.6613.137 | 53 | 29 | 56 | 1722 | 13308 | 16279 |
| Chrome-129.0.6668.58 | 54 | 30 | 58 | 1724 | 13310 | 16422 |
| Chrome-130.0.6710.0 | 56 | 32 | 61 | 1726 | 13332 | **16473** |
| HeadlessChrome-113.0.5672.63 | 37 | 12 | 40 | 1725 | **12334** | 15380 |
| HeadlessChrome-114.0.5735.133 | 37 | 12 | 42 | 1725 | 12396 | 15409 |
| HeadlessChrome-115.0.5790.170 | 37 | 12 | 40 | 1725 | 12500 | 15510 |
| HeadlessChrome-116.0.5845.96 | 37 | 12 | 41 | 1725 | 12522 | 15521 |
| HeadlessChrome-117.0.5938.149 | 36 | 14 | 40 | 1725 | 12540 | 15518 |
| HeadlessChrome-118.0.5993.70 | 36 | 15 | 41 | 1725 | 12550 | 15567 |
| HeadlessChrome-119.0.6045.105 | 35 | 15 | 40 | 1724 | 12558 | 15589 |
| HeadlessChrome-120.0.6099.109 | 35 | 15 | 40 | 1725 | 12588 | 15433 |
| HeadlessChrome-121.0.6167.184 | 35 | 15 | 40 | 1725 | 12608 | 15472 |
| HeadlessChrome-122.0.6261.128 | 34 | 15 | 39 | 1724 | 12607 | 15540 |
| HeadlessChrome-123.0.6312.122 | 33 | 15 | 39 | 1725 | 12650 | 15616 |
| HeadlessChrome-124.0.6367.207 | 32 | 15 | 37 | 1604 | 12676 | 15623 |
| HeadlessChrome-125.0.6422.141 | 33 | 14 | 38 | 1604 | 12757 | 15669 |
| HeadlessChrome-126.0.6478.182 | 33 | 14 | 37 | 1723 | 12779 | 15722 |
| HeadlessChrome-127.0.6533.119 | 34 | 14 | 37 | 1722 | 12785 | 15748 |
| HeadlessChrome-128.0.6613.137 | 41 | 15 | 46 | 1696 | 13308 | 16279 |
| HeadlessChrome-129.0.6668.58 | 41 | 15 | 46 | 1696 | 13310 | 16422 |
| HeadlessChrome-130.0.6710.0 | 41 | 15 | 47 | 1694 | 13332 | **16473** |

versions allowed us to exhaustively measure and comprehend the impact of each configuration parameter on the BOM.

As mentioned in the official Chromium documentation, the configuration for switches can be found on the *peter.sh* blog[7]. We tested a total of $1,748$ switches. Throughout our tests, an average of $1,710$ switches were successfully executed in the Headless browser version without encountering a timeout (3 timeouts were reported on each version) or a crash (23 to 144 crashed depending on version), and an average of $1,714$ switches in the graphical version (22 to 38 crashes). Out of all launched switches, 12 to 15 switches affected the fingerprint from the FingerprintJS library, and 32 to 41 switches impacted the fingerprint from the BOM exploration in the headless Chrome version. In the Headful version, we observed 15 to 32 switches affecting the FingerprintJS, and 40 to 56 switches affecting the BOM exploration. The differences between each Headless or Headful version is due to the versions themselves, as we utilized all existing switches from the latest version, and some of these switches did not exist in earlier versions. However, the difference between Headless and Headful browser versions lies in the additional functions for graphical processing in the latter. This outcome opens the possibility of foreseeing a bot detection system designed for bots that operate without utilizing the graphical interface.

The majority of impacted fingerprints from FingerprintJS were also impacted in the BOM exploration, except for the switches that affected the canvas generation. Switches like `disable-reading-from-canvas`, `force-high-contrast`, and

---

[7]https://peter.sh/experiments/chromium-command-line-switches/

force-prefers-reduced-motion were detected with the FingerprintJS library. The switch `disable-font-subpixel-positioning` was detected in the Headless Chromium version only, showing that certain configuration parameters also influence the canvas and can therefore be detected. The switch `headless` applied in Headful version shows that it impacts the browser fingerprint, and is therefore detectable on all the versions we tested. Some attributes can be affected by different switches, such as those ending with `length`, which are regularly impacted. The most affected attribute was `window._length`, which was impacted by $\approx 22$ different switches followed by `window.wgl_length` impacted by $\approx 8$ different switches.

As explained earlier, some attributes can be impacted by different switches. However, in the majority of cases, each switch impacts the value of the attribute similarly, except for some attributes like `length` ones for which values can vary substantially. The limited number of collisions and relative independence allows us to avoid exploring all possible combinations of switches, which would be unfeasible given the number of switches, and instead create our dataset by enabling switches linearly, one at a time.

## 5.2 Configuration Identification [RQ2]

After evaluating the impact of each configuration parameter, FP-Rainbow leverages this information to identify an unknown fingerprint. The effectiveness of FP-Rainbow is evaluated in two stages: *(i)* identifying the browser fingerprints impacted by the configuration parameters in the dataset and *(ii)* generating and identifying browser fingerprints with several switches activated at the same time.

*5.2.1 Baseline identification.* First, we attempt to identify switches from fingerprints that already exist in our dataset from the data collection and analysis processes. This provides a baseline to understand if a switch is identifiable. As shown in Table 2, FP-Rainbow successfully identified the majority of the switches. During the experimentation, we observed that some switches enable the same feature and have exactly the same impact on the BOM, such as `disable-3d-apis` and `disable-webgl`. After verifying the source code[8], we can confirm they are perfectly identical. We consider such switches as *equivalent* switches. The dataset also contains cases where a switch is a perfect subset of another. This includes `disable-speech-synthesis-api` being a part of `disable-speech-api`. In some cases, a switch like `disable-3d-apis` can have both an equivalent and a subset like `disable-gpu-driver-bug-workarounds`. Identifying subsets allows us to distinguish if the child switch is enabled and not the parent switch. Table 2 also reveals that Chrome versions 128.0.6613.137, 129.0.6668.58 and 130.0.6710.0 exhibit a significantly higher number of impacted switches compared to earlier versions.

*5.2.2 Multi-switch identification.* We repeated the same experiment described in Section 5.2.1 but with multiple switches. Since it is impractical to explore every possible combination and there is no statistical data on the number of switches users typically activate,

we devised a random sampling strategy to obtain representative samples. We generated 31 fingerprints per browser version. To cover as many browser fingerprints as possible, we used a pool of switches, distributing them evenly across tests. Specifically, we generated 10 browser fingerprints using $\frac{1}{4}$ of the available switches, then repeated the process with $\frac{1}{2}$ of the switches and $\frac{3}{4}$ of the switches. Finally, we generated a fingerprint incorporating all the switches impacted by this version. The result was a successful identification rate of 78.15% across all fingerprints with multiple switches, including identifying various switch combinations across all tested versions, as shown in Table 2.

## 5.3 Leveraging Single-Device Dataset for Fingerprint Identification Across Multiple Devices [RQ3]

We repeat the experiment reported in Section 5.2.2 that was run on a Server, but instead collect the fingerprints from a different environment running on a Laptop. The setup is otherwise the same as in Section 4.2. As shown in Table 2, some identification results are identical or very similar for fingerprints generated on the two different machines. However, the differences between the two experiments show that changing the environment impacts the BOM. On average, FP-Rainbow achieves a recognition rate of 77.54%, compared to 78.15%, when testing fingerprints from a different environment, showing we can leverage single-device datasets to a large extent. Despite technical issues with older versions (113 and 114) which are not shown in the Table, the results indicate that FP-Rainbow can successfully identify configuration parameters in environments other than the original dataset.

## 5.4 Discussion

*Minimal Fingerprint.* Using FP-Rainbow, it is possible to determine a minimal browser fingerprint for the identification of configuration parameters. To find out this minimal fingerprint, proper attributes must be selected. To do this, the *Stability* formula (see formula 2) is applied when aiming for one specific version, or the $\omega_a$ formula (see formula 3) to broaden the scope over all versions. Attributes are then ranked based on stability. If an attribute is consistently linked to a single switch across all versions, it is considered stable and ranked higher. Conversely, if an attribute is impacted by multiple switches and/or only on certain versions, it is considered unstable and ranked lower. A threshold is then set for selecting the number of attributes, either regarding performance or precision. Selecting more attributes increases identification accuracy, but also extends the time required for generating and processing the fingerprint. This approach allows developers to inform users of suboptimal or inadequate configurations and suggest improvements when visiting the website.

*Limitations.* As previously highlighted, the attributes impacted by switches may sometimes overlap with another switch and reduce the identification rate, such as with the `disable-webgl` and `disable-3d-apis` switches (see Section 5.2). Additionally, we noticed some noise during our analysis, like unstable attributes, or attributes that have been impacted outside of our configuration parameters. This noise can be mitigated through an in-depth analysis

---

[8]https://source.chromium.org/chromium/chromium/src/+/main:
content/browser/web_contents/web_contents_impl.cc;l=3000;drc=
7fa0c25da15ae39bbd2fd720832ec4df4fee705a

**Table 2: Analysis of the effects of switches on the BOM and switch identification.** *Left* provides statistics on the generation of browser fingerprints, attribute density for specific browser versions, results of individual switch identification from the BOM, equivalent and subset switches [RQ1]. *Middle-right* shows the results of the multi-switch identification experiment [RQ2] on the same platform that collected the dataset (Server), and *Right* shows the multi-switch identification platform with fingerprints from a different platform [RQ3].

| | Impact [RQ1] and Identification [RQ2] of switches from browser fingerprints | | | | | Multiswitch identification experiment Server [RQ2] | | | | Multiswitch identification experiment Laptop [RQ3] | | | |
|---|---|---|---|---|---|---|---|---|---|---|---|---|---|
| Browsers version | Switches that impacted the BOM | Identified switches | Failed switch identification | Equivalent switches | Subset switches | Number of switches tested | Identified switches | Mistaken identification (False Positive) | Unidentified (False Negative) | Number of switches | Identified switches | Mistaken identification (False Positive) | Unidentified (False Negative) |
| Chrome-113.0.5672.63 | 45 | 89.13% (41) | 5 | 9 | 6 | 461 | 75.9% (350) | 4.8% (22) | 24.1% (111) | | | | |
| Chrome-114.0.5735.133 | 46 | 89.36% (42) | 5 | 9 | 7 | 751 | 76.3% (573) | 2.9% (22) | 23.7% (178) | | | | |
| Chrome-115.0.5790.170 | 46 | 89.36% (42) | 5 | 9 | 7 | 751 | 77.0% (578) | 3.1% (23) | 23.0% (173) | 136 | 84.6% (115) | 8.1% (11) | 15.4% (21) |
| Chrome-116.0.5845.96 | 45 | 89.13% (41) | 5 | 9 | 6 | 742 | 76.8% (570) | 3.5% (26) | 23.2% (172) | 110 | 87.3% (96) | 10.0% (11) | 12.7% (14) |
| Chrome-117.0.5938.149 | 45 | 89.13% (41) | 5 | 9 | 8 | 742 | 79.5% (590) | 3.2% (24) | 20.5% (152) | 183 | 80.9% (148) | 5.5% (10) | 19.1% (35) |
| Chrome-118.0.5993.70 | 44 | 88.89% (40) | 5 | 9 | 7 | 720 | 79.2% (570) | 3.5% (25) | 20.8% (150) | 182 | 79.1% (144) | 6.6% (12) | 20.9% (38) |
| Chrome-119.0.6045.105 | 44 | 88.89% (40) | 5 | 9 | 6 | 719 | 80.7% (580) | 2.9% (21) | 19.3% (139) | 719 | 80.3% (577) | 3.2% (23) | 19.7% (142) |
| Chrome-120.0.6099.109 | 42 | 88.37% (38) | 5 | 8 | 5 | 687 | 82.0% (563) | 3.2% (22) | 18.0% (124) | 688 | 81.8% (563) | 3.2% (22) | 18.2% (125) |
| Chrome-121.0.6167.184 | 42 | 88.37% (38) | 5 | 8 | 5 | 689 | 82.6% (569) | 3.6% (25) | 17.4% (120) | 612 | 82.5% (505) | 3.9% (24) | 17.5% (107) |
| Chrome-122.0.6261.128 | 41 | 88.10% (37) | 5 | 8 | 5 | 677 | 81.8% (554) | 3.4% (23) | 18.2% (123) | 677 | 81.8% (554) | 3.4% (23) | 18.2% (123) |
| Chrome-123.0.6312.122 | 42 | 88.10% (37) | 6 | 8 | 7 | 676 | 80.8% (546) | 3.8% (26) | 19.2% (130) | 634 | 80.4% (510) | 4.1% (26) | 19.6% (124) |
| Chrome-124.0.6367.207 | 41 | 87.80% (36) | 4 | 8 | 6 | 657 | 80.8% (531) | 3.2% (21) | 19.2% (126) | 615 | 80.7% (496) | 3.7% (23) | 19.3% (119) |
| Chrome-125.0.6422.141 | 40 | 87.50% (35) | 6 | 8 | 6 | 656 | 80.5% (528) | 4.3% (28) | 19.5% (128) | 656 | 80.5% (528) | 4.3% (28) | 19.5% (128) |
| Chrome-126.0.6478.182 | 40 | 87.50% (35) | 6 | 8 | 6 | 655 | 80.9% (530) | 4.1% (27) | 19.1% (125) | 655 | 80.9% (530) | 4.1% (27) | 19.1% (125) |
| Chrome-127.0.6533.119 | 40 | 87.50% (35) | 6 | 8 | 6 | 656 | 80.5% (528) | 0.9% (6) | 19.5% (128) | 656 | 80.5% (528) | 0.9% (6) | 19.5% (128) |
| Chrome-128.0.6613.137 | 53 | 67.92% (36) | 18 | 8 | 6 | 849 | 63.4% (538) | 2.5% (21) | 36.6% (311) | 849 | 63.4% (538) | 2.5% (21) | 36.6% (311) |
| Chrome-129.0.6668.58 | 54 | 66.67% (36) | 19 | 8 | 5 | 868 | 62.3% (541) | 4.6% (40) | 37.7% (327) | 868 | 62.3% (541) | 4.6% (40) | 37.7% (327) |
| Chrome-130.0.6710.0 | 56 | 64.29% (36) | 21 | 8 | 6 | 911 | 59.5% (542) | 2.9% (26) | 40.5% (369) | 911 | 59.5% (542) | 2.9% (26) | 40.5% (369) |
| HeadlessChrome-113.0.5672.63 | 37 | 86.84% (33) | 5 | 6 | 6 | 103 | 79.6% (82) | 7.8% (8) | 20.4% (21) | | | | |
| HeadlessChrome-114.0.5735.133 | 37 | 86.84% (33) | 5 | 6 | 6 | 613 | 79.4% (487) | 4.7% (29) | 20.6% (126) | | | | |
| HeadlessChrome-115.0.5790.170 | 37 | 86.84% (33) | 5 | 6 | 6 | 613 | 80.3% (492) | 4.1% (25) | 19.7% (121) | 614 | 77.9% (478) | 6.7% (41) | 22.1% (136) |
| HeadlessChrome-116.0.5845.96 | 37 | 86.84% (33) | 5 | 6 | 6 | 611 | 80.5% (492) | 4.1% (25) | 19.5% (119) | 611 | 77.7% (475) | 7.0% (43) | 22.3% (136) |
| HeadlessChrome-117.0.5938.149 | 36 | 86.49% (32) | 5 | 6 | 6 | 590 | 80.3% (474) | 4.1% (24) | 19.7% (116) | 591 | 77.8% (460) | 9.5% (56) | 22.2% (131) |
| HeadlessChrome-118.0.5993.70 | 36 | 83.78% (31) | 6 | 6 | 5 | 591 | 79.7% (471) | 6.6% (39) | 20.3% (120) | 591 | 76.6% (453) | 12.2% (72) | 23.4% (138) |
| HeadlessChrome-119.0.6045.105 | 35 | 86.11% (31) | 5 | 6 | 4 | 591 | 79.0% (467) | 4.1% (24) | 21.0% (124) | 591 | 77.3% (457) | 12.2% (72) | 22.7% (134) |
| HeadlessChrome-120.0.6099.109 | 35 | 86.11% (31) | 5 | 6 | 4 | 588 | 81.6% (480) | 4.9% (29) | 18.4% (108) | 588 | 79.4% (467) | 12.2% (76) | 20.6% (121) |
| HeadlessChrome-121.0.6167.184 | 35 | 86.11% (31) | 5 | 6 | 4 | 590 | 81.5% (481) | 4.1% (24) | 18.5% (109) | 591 | 79.4% (469) | 12.5% (74) | 20.6% (122) |
| HeadlessChrome-122.0.6261.128 | 34 | 85.71% (30) | 5 | 6 | 4 | 559 | 81.4% (455) | 4.8% (27) | 18.6% (104) | 559 | 78.5% (439) | 13.6% (76) | 21.5% (120) |
| HeadlessChrome-123.0.6312.122 | 33 | 85.29% (29) | 5 | 6 | 5 | 547 | 80.3% (439) | 4.4% (24) | 19.7% (108) | 547 | 77.9% (426) | 13.5% (74) | 22.1% (121) |
| HeadlessChrome-124.0.6367.207 | 32 | 87.88% (29) | 4 | 6 | 5 | 529 | 82.4% (436) | 6.0% (32) | 17.6% (93) | 529 | 79.4% (420) | 14.2% (75) | 20.6% (109) |
| HeadlessChrome-125.0.6422.141 | 33 | 88.24% (30) | 4 | 6 | 5 | 550 | 82.5% (454) | 4.2% (23) | 17.5% (96) | 550 | 78.2% (430) | 13.3% (73) | 21.8% (120) |
| HeadlessChrome-126.0.6478.182 | 33 | 85.29% (29) | 5 | 6 | 5 | 549 | 80.1% (440) | 3.8% (21) | 19.9% (109) | 549 | 74.7% (410) | 10.6% (58) | 25.3% (139) |
| HeadlessChrome-127.0.6533.119 | 34 | 85.71% (30) | 5 | 6 | 5 | 558 | 79.4% (443) | 4.3% (24) | 20.6% (115) | 558 | 76.2% (425) | 10.9% (61) | 23.8% (133) |
| HeadlessChrome-128.0.6613.137 | 41 | 80.49% (33) | 9 | 5 | 6 | 656 | 76.1% (499) | 4.7% (31) | 23.9% (157) | 656 | 75.0% (492) | 5.5% (36) | 25.0% (164) |
| HeadlessChrome-129.0.6668.58 | 41 | 78.05% (32) | 9 | 5 | 4 | 656 | 74.7% (490) | 6.7% (44) | 25.3% (166) | 656 | 74.7% (490) | 6.6% (43) | 25.3% (166) |
| HeadlessChrome-130.0.6710.0 | 41 | 78.05% (32) | 9 | 5 | 4 | 657 | 74.1% (487) | 7.2% (47) | 25.9% (170) | 657 | 74.1% (487) | 7.0% (46) | 25.9% (170) |

of each attribute, which may lead to the exclusion of such attributes during the identification phase or the assignation of a lesser weight due to their instability or randomness.

Several factors introduce potential threats to the validity of our study. First, our dataset was exclusively generated using Headless and Headful Chromium versions within a Docker environment, which may restrict its generalizability to real-world scenarios. Additionally, the absence of datasets generated on Windows or Mac platforms, as well as the omission of scenarios involving a GPU or a genuine graphical interface, could limit the general applicability of our findings.

## 6 CONCLUSION

This paper introduced FP-Rainbow, a systematic approach to investigate and identify browser settings and configurations influencing the BOM. We explore over 1,700 configuration parameters and identify 61 of which impact the Browser Object Model (BOM) and its fingerprint as seen through FingerprintJS. We show that it is possible to identify configuration parameters from unknown fingerprints, albeit some parameters being unstable are not always properly identified. We show that altering a browser's configuration impacts the attack surface, allowing for possible attacks or improved fingerprint tracking approaches, and should be of concern to both users as well as developers. This work opens up several avenues for future research. In particular, we plan to investigate additional factors that influence the browser's fingerprint, such as user settings, hardware, firmware, plugins, and extensions. We expect to improve our tooling to better inform developers of the side-effects of the features they add or the changes they introduce into browsers.

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
