# OpenReview forum: "FP-Rainbow : Fingerprint-based Browser Configuration Identification"
_ACM.org/TheWebConf/2025/Conference — WWW 2025 Poster_

### Official Review · Reviewer_e43e · 2024-11-20

**Novelty:** 6
**Technical Quality:** 4

**Review:**

The paper presents an interesting study on the impact of browser configuration parameters on its fingerprint, and how to infer those parameters given a fingerprint. This is an important area of research as different browser versions and configurations might suffer from security vulnerabilities which might be exploited by malicious actors.


**Originality**

The has been very little related work, which the authors have references and pointed out the ways in which their work is different and the contributions it makes to the research area. No concerns on this front.

**Clarity**
The paper is well written and easy to read. The structure is sound and the figures/tables are clear and legible. No concerns on this front.

**Quality**
The methodology of this study seems sound, but it tackles the problem in a somewhat limited form, in addition to lacking in depth and details at times. The main issues are as follows:
* The study examined in detail the fingerprints by changing only one parameter at a time. Although this makes it easy to experiment with, it begs the question of how realistic and how often this happens in reality. Although this is an interesting academic exercise, the paper should have expanded on the evaluation of multi-switch fingerprints. The authors mention this in Section 5.2.2 and provide some results in Table 2, but these need to be significantly expanded and contrasted with the single-parameter case.
* The study only considered 2 hardware platforms (a VM server and a laptop running Linux). No Windows, Mac OS or Chrome OS platform was considered. This greatly limits the significance of the results as Linux has a ~4% market share.

_Additional comments_
* The authors should clarify what is a fingerprint in their case. As far as I understood, it is not just an actual string that is derived by hashing the values of different API return values, but it is rather a set of strings, each one corresponding to one specific return value of the called APIs. For instance, the authors state that they used the FingerprintJS library to look at the APIs, and therefore should clearly state which APIs exactly were used and what was stored (I assume the return values individually, not just the single string that is returned by the FingerprintJS call for a fingerprint).
* There should be a glossary of the different variable names used in the equations, as there are many. Moreover, these names should be shortened as they are somewhat hard to parse when they are long (e.g., Add_a,impactedSubset_v,unk)
* l. 616 - 622: Unclear what the diff is.
* l. 685 - 687: Are these the only differences between headful and headless? Did you look for other APIs typically available on a headful/user machine (e.g,., network card, audio) but not necessarily in headless mode?
* l. 710 716: This requires a proper detailed analysis (i.e., in the majority of cases, each switch impacts the value of the attribute similarly, except for some attributes ..."
* l. 790.795: I was not able to find the top-N attributes anywhere. Please include them.

**Questions:**

* Would like to see a more detailed analysis and comparison between single-parameter change and multi-parameter re-identification performance.
* Headful vs headless, more detailed explanation and analysis of differences.
* More hardware platforms

**Reviewer Confidence:**

3: The reviewer is confident but not certain that the evaluation is correct

**Scope:**

4: The work is relevant to the Web and to the track, and is of broad interest to the community

---

### Official Review · Reviewer_wEU1 · 2024-11-30

**Novelty:** 3
**Technical Quality:** 4

**Review:**

This paper proposes FP-Rainbow, a novel approach that systematically explores the mapping relation between Chromium-based web browser fingerprintings and configuration parameters. FP-Rainbow collects and analyzes thousands of fingerprints from various versions of Chromium to reveal the influence relationship and achieve unknown fingerprinting identification.


**Pros:**

1. First to conduct the relationship between configuration parameters and web browser fingerprints, and further realize unknown fingerprint identification.
2. Provides a large-scale fingerprints dataset for Chromium-based web browser.
3. Helps browser developers better understand the impact of configuration parameters on user privacy, leading to the development of safer browsers.


**Cons:**

1. The paper is hard to follow. The background knowledge and technical details are oversimplified, which may not be sufficient for understanding the FP-Rainbow method. For example, the definition of attributes and switches in key contributions in the Introduction does not appear in the previous section.
2. **[Severe]** The dataset used in this paper was generated in a Docker environment and may not fully reflect real-world browser behavior. As a measurement research, whether the experimental results in this paper can reflect and portray the situation in real-world environments is of great significance in evaluating its value. Although the authors have mentioned the limitations of using virtualized environments in the Discussion, neither the further description nor the experimental evaluation that quantifies possible impact is provided. This is very concerning as readers are unable to assess the usability and correctness of the generated fingerprints and their relationship with configuration parameters in real-world environments.
3. The experiments do not fully consider existing anti-fingerprinting tools, which may be involved in inadequate comparisons. A detailed evaluation of FP-Rainbow under existing anti-fingerprinting methods is necessary.

**Questions:**

Please refer to the **Cons** above.

**Reviewer Confidence:**

3: The reviewer is confident but not certain that the evaluation is correct

**Scope:**

3: The work is somewhat relevant to the Web and to the track, and is of narrow interest to a sub-community

---

### Official Review · Reviewer_maYm · 2024-12-01

**Novelty:** 5
**Technical Quality:** 5

**Review:**

### Summary

This paper explores and measures how a browser’s configuration parameters (switches) influence its fingerprint, known as the Browser Object Model (BOM). The authors collect real-world browser fingerprints and configuration parameters from various versions of Chromium to create a dataset. They then identify key configuration parameters that influence the fingerprint, demonstrate how to use the fingerprint to re-identify browser configuration parameters, and explain how to effectively collect fingerprints from a single device that can indicate fingerprints across multiple devices. The accuracy rate for re-identifying configuration parameters from fingerprints is at least 78%.

### Motivation and focus

The motivation for this work is valuable, and the problem itself is significant. Browser fingerprinting is a well-defined area, but what is less known is how configurations impact a browser’s fingerprint and how configuration parameters can be inferred from a given fingerprint.

Although prior work [29] illustrates that attributes can be exploited by attackers, the influence of configuration parameters (i.e., switches) on security is not immediately apparent. Therefore, additional works could be cited to strengthen the motivation for RQ2.

### Technical Correctness

The paper provides experiments to analyze the impact of switches by identifying differences in fingerprints. However, the question of which specific switches or configurations can lead to severe consequences remains unanswered. Adding more discussions on this aspect would improve the paper.

While the identification of a single switch is understandable, identifying multiple switches (as mentioned in subsection 5.2) is less clear. The authors randomly selected some of the switches to change and generated the corresponding fingerprints. However, the paper only presents the result (recognition rate) without explaining how this identification was performed. Additionally, it remains unclear what types of combinations lead to changes in fingerprints. For instance, if switches A and B together influence a fingerprint, but neither A nor B individually has any impact, how would this case be detected?

What is the time cost of generating a large fingerprint, and what are its influences? Subsection 5.4 discusses the minimal fingerprint but does not address what fingerprint size would be appropriate. Additionally, the paper does not provide information on the typical size of fingerprints generated by FingerprintJS.

Additionally, the authors identify 61 switches that impact the fingerprints. However, the reasons why these switches have such an impact could be analyzed and discussed in greater depth.

### Related work

The related work is adequate.

### Presentation

In Sections 3 and 4, the descriptions are overly high-level and confusing. They lack details about what a fingerprint looks like, how it is mapped to configuration parameters, and what the attributes are. I did not fully understand the overall concept until diving into the details in Section 5. This could be significantly improved by including more graphs or a motivating example.

The paper primarily focuses on how switches impact fingerprints, but it does not provide sufficient background or descriptions about the switches, which adds to the confusion. Additionally, it is unclear how to modify a configuration parameter to observe its impacts on the fingerprints.

**Questions:**

1. The authors should clarify the benefits of identifying the related switches from a given fingerprint. What vulnerabilities could be exploited in such scenarios?
2. Please provide detailed clarification on how to detect the combination of switches mentioned earlier. Additionally, what are the time costs associated with achieving the identification rate?
3. How many attributes does FingerprintJS generate? Could a large fingerprint introduce performance issues?

**Reviewer Confidence:**

3: The reviewer is confident but not certain that the evaluation is correct

**Scope:**

4: The work is relevant to the Web and to the track, and is of broad interest to the community

---

### Official Review · Reviewer_1EG3 · 2024-12-02

**Novelty:** 4
**Technical Quality:** 5

**Review:**

This paper presents FP-Rainbow, a systematic approach to explore and map the configuration space of Chromium-based web browsers to identify the impact of configuration parameters on the Browser Fingerprint. The researchers analyzed 1,748 configuration parameters and found that 37-61 parameters (of different browser versions) affect the browser's BOM. By collecting and analyzing over 61,000 fingerprint samples from 18 different versions of Chromium, the study revealed the impact of specific configuration parameters on browser fingerprints.
FP-Rainbow also demonstrated the ability to identify configuration parameters from unknown fingerprints, with an average accuracy of 84% when considering browser fingerprints and 78% when multiswitch server and laptop identification are involved. These results emphasise the importance of measuring the impact of configuration parameters on browsers in order to develop more secure and privacy-conscious web browsers.

Pros:
1. The study examined 1,748 configuration parameters and determined that between 32 and 56 of them (depending on the version) have an impact on the BOM. This comprehensive coverage helps reveal which settings are particularly important.
2. The FP-Rainbow method is effective in identifying configuration parameters from unknown browser fingerprints, with an average successful identification rate of 84% for switches from browser fingerprints and 78% for multi switch identification experiment. This demonstrates the high accuracy of the method even in the face of unknown or complex situations.
3. By implementing experiments using 1,116 randomly sampled browser configurations, it is demonstrated that a dataset computed on one device can be used to successfully identify the configuration parameters on other devices. This means that the need to generate datasets on a number of different devices is reduced, thus increasing efficiency.

Cons:
1. Different browser versions have a significant effect on fingerprint generation. Maintaining the accuracy of the analysis may require model updates if future browser versions are updated. This may mean that a significant amount of time and analytical work will be required for each major browser update.
2. The study was only explored for the Chromium browser. It is uncertain whether similar results could be obtained if FP-Rainbow were to be extended to other browsers, suggesting that the scalability of the methodology is questionable
3. The model focuses primarily on Chromium and its derivative browsers and may not be fully applicable to other browser engines. It is also necessary to continuously monitor changes in configuration parameters over time to assess their impact on browser fingerprinting over time. Future work should consider the impact of additional factors such as user settings, hardware, firmware, plug-ins, and extensions on browser fingerprints.
4. The researchers used a systematic approach to explore the impact of configuration parameters on browser fingerprinting, but the dataset was generated in a Docker environment using Headless and Headful Chromium versions, which may limit the general applicability of the findings, especially in the absence of coverage of Windows or Mac platforms
5. Although FP-Rainbow demonstrates an effective method for identifying configuration parameters from unknown fingerprints, it does not provide sufficient evidence that this method is superior to existing solutions in the literature in all cases.

**Questions:**

1. According to the description of the paper, different browser versions can have a significant impact on fingerprint generation. If future browser versions are updated, can you still maintain the accuracy of your analysis without updating the model? If needed, will it still require a lot of time and analysis as described in the paper?
2. The paper only studied the Chromium browser. If FP-Rainbow is extended to other browsers, can similar results be achieved? In other words, how scalable is FP-Rainbow?
3. The dataset was generated exclusively using Headless and Headful Chromium versions within a Docker environment. Is it compatible with Chromium-based applications?
4. Is it possible to identify a configuration parameter from an unknown fingerprint of Chromium-based applications?
5. The study hypothesizes that different configuration parameters can significantly change the fingerprint of the browser, which is reasonable in practical applications. This is because configuration parameters directly affect the behavior and functionality of the browser, thus changing the information it presents to the external environment. However, the model has some limitations; e.g., it focuses on Chromium and its derivative browsers and may not be fully applicable to other browser engines. However, we continuously monitor changes in configuration parameters over time to assess their long-term impact on browser fingerprinting. Future work should also consider additional factors such as user settings, hardware, firmware, plug-ins, and extensions that may affect browser fingerprints.
6. The researchers used a systematic approach to explore the effects of configuration parameters on browser fingerprinting. They not only collected a large amount of data, but also verified the consistency and stability of the results by comparing the differences between different versions. In addition, they considered differences in the way different types of data (strings, numeric values, PNG images) are handled. However the dataset was generated in a Docker environment using Headless and Headful Chromium versions. Although this approach helps to control variables, it may limit the generalizability of the findings, especially if Windows or Mac platforms are not covered. Therefore, the choice of dataset is appropriate for the study objectives, but may need to be extended further for wider application scenarios.
7. Although FP-Rainbow demonstrates an effective method for identifying configuration parameters from unknown fingerprints, it does not provide sufficient evidence that this method outperforms existing solutions in all cases when compared to the existing literature.
8. No test cases involving the GPU or real graphical interfaces were included, which may have overlooked the impact of the GPU on browser fingerprinting, especially when working with images and video content.

**Reviewer Confidence:**

3: The reviewer is confident but not certain that the evaluation is correct

**Scope:**

4: The work is relevant to the Web and to the track, and is of broad interest to the community